# Pulsed Plasma Accelerator

Alexander Karimov [1,2,*], Svyatoslav Terekhov [2] and Vladimir Yamschikov [3]

1 Institute for High Temperatures, Russian Academy of Sciences, Izhorskaya 13/19, 127412 Moscow, Russia
2 Department of Electrophysical Facilities, National Research Nuclear University MEPhI, Kashirskoye Hway 31, 115409 Moscow, Russia
3 Institute for Electrophysics and Electric Power, Russian Academy of Sciences, Dvortsovaya Naberezhnaya 18, 191186 Saint-Petersburg, Russia
* Correspondence: arkarimov@mephi.ru

**Abstract:** In this paper, we consider the acceleration of plasma fluxes in crossed electromagnetic fields. The possible technical approach to a prospective plasma accelerator is discussed. A simple hydrodynamic model describing the dynamics of the plasma ring in these fields is proposed. Based on this model, the estimations of basic characteristics for the accelerated flux are calculated for typical experimental conditions.

**Keywords:** plasma accelerator; pulsed magnetic field; ponderomotive force; magnetic circuit; plasma conductivity





## 1. Introduction

Creating new methods and devices for the collective acceleration of plasma fluxes is of interest for different technical applications including space exploration, especially the design of propulsion implications of spacecraft [1–6]. In fact, the thrust caused by an electric propulsion system is only limited by the amount of power that can be supplied to the device. A greater efficiency of plasma thrusters can be achieved through an acceleration of plasma flow which may appear in quasi-neutral conditions when there is no limitation brought about by the space charge of the flow.

Among several existing ways for the technical implementation of such plasma acceleration, the Hall accelerators appear to be potentially the most promising. In this kind of thruster, the orthogonal external electric $E_0$ and magnetic $B_0$ fields are used to accelerate the plasma flow. The radial magnetic field is assumed to be large enough to magnetize electrons but not ions. As a result, the electrons can experience $B_0 \times E_0$ rotation, and ions may move only in the axial direction. That is, a radial magnetic field shall confine electrons which compensate for the space charge of the flow, while an axial electric field shall accelerate ions. However, the effect of charge polarization arising in such a system has to restrict the acceleration process. One can go to the limits of the magnetized electron approximation if both ion and electron flows move in one direction. Such acceleration may occur if plasma flow is present in the crossed fields varied with time and space; the typical technical realization of such an idea is as an end-Hall thruster [1,7].

This idea has been developed in [8,9] where we have considered the dynamics of different plasma flows in crossed magnetic fields taking into account the internal fields in plasma. In these works, we have left out the question about the technical realization of the approaches proposed as well as the problem of the influence of external fields on the dynamics and structure of the plasma bunch. Proceeding from our previous results, in the present paper, we touch upon these points.

## 2. Operational Principle and Design of Pulsed Accelerator

Here we shall consider a thruster concept that is a hybrid of induction and end coaxial plasma accelerators [8,9]. A possible technical implementation of such an idea shown in

Figure 1 [10] was partially implemented in the experiments [11] where the plasma bunch in the form of a ring (see Figure 2) consisting of electrons and light positive ions was generated as a result of discharge over a dielectric surface.

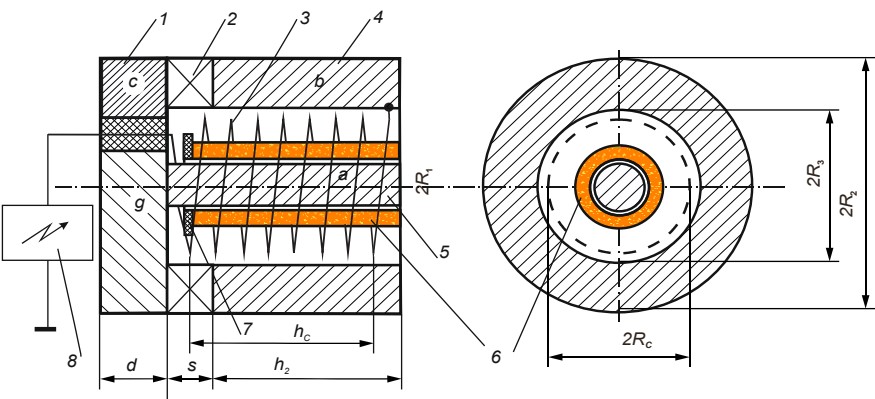

**Figure 1.** The accelerator schematic [10]: 1—disk-shaped magnetic circuit; 2—permanent magnet; 3—spiral electrode; 4—tubular magnetic circuit; 5—cylindrical magnetic circuit; 6—accelerated plasma flow; 7—dielectric washer; 8—current pulse generator.

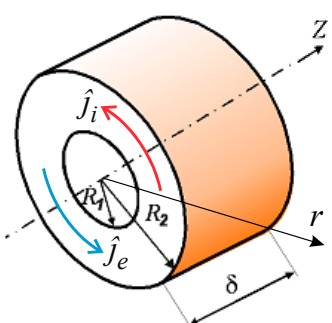

**Figure 2.** The plasma ring in the cylindrical coordinate system which is accelerated in the device shown in Figure 1. Here, $R_1$ and $R_2$ are the inner and the external radii of the plasma ring; $\delta$ is the ring length.

It should be noted that this plasma ring in the present scheme can also be obtained in other ways. Generally speaking, the question of plasma generation requires a separate discussion and we hope to return to this point in our forthcoming works. We shall consider that the acceleration of the plasma bunch is caused by the ponderomotive force that arises in an axisymmetric magnetic field:

$$\boldsymbol{B}_0 = B_{r0}\boldsymbol{e}_r + B_{z0}\boldsymbol{e}_z, \tag{1}$$

where $B_{r0}$ and $B_{z0}(t)$ are the stationary radial and time-dependent longitudinal components of the magnetic induction vector. As can be seen in Figure 1, a permanent magnetic field is formed in the space between the tubular and cylindrical magnetic circuits. Setting the geometric factor $\chi = (R_2 - R_1)/h_2 \ll 1$, we may neglect the scattering fluxes of the magnetic field.

Herewith, the axial component of the constant magnetic field in the gap is assumed to be equal to zero, and the radial component shall be determined by the following expression:

$$B_{r0} \approx B_m \frac{R_3^2 - R_2^2}{2h_2 r} = B_m \frac{L}{r}, \tag{2}$$

where $B_m$ is the magnetic field induction at the end face of the annular permanent magnet, and the value $L = (R_3^2 - R_2^2)/2h_2$ plays the role of the natural spatial scale for the problem worked out. For geometric clarity, here we set $L = R_1$. The nonstationary axial component $B_{z0}$ is given by

$$B_{z0} = B_s \begin{cases} \mu, & 0 \leq r < R_1 \\ 1, & R_1 \leq r \leq R_c \end{cases} \tag{3}$$

where $\mu$ is the relative magnetic permeability of the magnetic core material, and $B_s \approx 4\pi NI/ch_c$; here, $N$ is the number of turns of the helix, $h_c$ is the length of the helix, $I = I(t)$ is the current in the helix, and $c$ is the speed of light in vacuum. (Here and henceforth, all physical quantities are written in the Gaussian unit system).

Such a nonstationary magnetic field $B_{z0}$ repeats the temporal form of the signal $I(t)$ and generates an azimuthal electric field

$$E_\theta = -\frac{1}{cr}\frac{d}{dt}\int_0^r r' B_{z0}(t, r', z)\,dr', \tag{4}$$

i.e., in the case under consideration, the external electric field is $\mathbf{E}_0 = E_\theta \mathbf{e}_\theta$. This field rotates the azimuthal electron and ion current fluxes in different directions. The interaction of these electron and ion fluxes with the radial magnetic field $B_{r0}$ leads to the acceleration of the entire plasma bunch in the axial z-direction. However, as is seen from (4), the magnetic field (3) must fulfill certain requirements in order to produce an accelerating effect. For example, for a harmonic function, $B_{z0}(t)$, the plasma acceleration can occur only for a quarter period. This implies that we have to use some electronic unit to form the signal $B_{z0}(t)$ waveform with a required phase. Generally, one can say that there exists an acceleration effect in some temporal interval of duration $t_i$ and then the signal $I(t)$ must be repeated. Namely, this property determines the impulse nature of the proposed accelerator scheme.

Moreover, it should be noted that the acceleration effect depends on the ratio between $B_{r0}$ and $\partial_t B_{z0}$ but does not depend on the magnitude of $B_{z0}$. This means that there is a set of geometric and technical parameters which can help in obtaining the acceleration of the plasma ring for

$$B_s \ll B_m \tag{5}$$

Thus, there is no need to create a large current $I$ in the solenoid. For simplicity, in the present paper, we restrict our consideration to the case of condition (5) being fulfilled. However, even in this simplest case, the nonstationary and nonlinear nature of processes in the plasma ring acceleration determines the complex character of the evolution of the system under consideration. In order to estimate this acceleration effect for the scheme worked out, we shall present a simple fluid model describing the dynamics of the plasma ring in crossed magnetic fields.

## 3. Model of Plasma Bunch

Neglecting the intrinsic electric and magnetic fields of the plasma bunch, as well as the inertia of electrons, we shall study the dynamics of the plasma ring formed only in external fields $\mathbf{E}_0$ and $\mathbf{B}_0$. In addition, assuming the electron and ion temperatures to be constant ($\nabla T_e = \nabla T_i = 0$), we restrict ourselves to studying a quasi-neutral plasma ring with electron and ion density $n_e = n_i = n$ in the azimuthally symmetric geometry ($\partial_\theta = 0$). In the general case, the evolution of the ring is described by the continuity equation for electrons ($s = e$) and ions ($s = i$):

$$\frac{\partial n}{\partial t} + \nabla \cdot (n \mathbf{v_s}) = 0, \tag{6}$$

as well as the equation of motion for ions in the form:

$$m_i n \frac{d\mathbf{v_i}}{dt} = en\left(\mathbf{E}_0 + \frac{1}{c}\mathbf{v_i} \times \mathbf{B}_0\right) - \frac{m_e \nu_{ei}}{e}\mathbf{j}, \tag{7}$$

where $m_e$ and $m_i$ are the masses of the electron and ion, respectively, and $\nu_{ei}$ is the frequency of electron–ion collisions, determined by [3]:

$$\nu_{ei} = \frac{\pi e^4 n_e}{m_e^{1/2} T_e^{3/2}} \ln(\Lambda)$$

where $\ln(\Lambda)$ is the Coulomb logarithm. For simplicity, we shall neglect the collisions of electrons with neutral particles, and bearing in mind relation (5), in all estimations, we shall use $B_0 \approx B_m$. Herewith, the current density $j = en(v_i - v_e)$ changes after the generalized Ohm's law [4,7]:

$$j = \sigma\left(E_0 + \frac{1}{c}v_i \times B_0\right) - \frac{\omega_{ce}}{\nu_{ei}}\frac{j \times B_0}{B_0}, \tag{8}$$

where $\sigma = e^2 n / m_e \nu_{ei}$ is the electronic conductivity and $\omega_{ce} = eB_0/cm_e$ is the electron cyclotron frequency estimated from $B_0$.

In addition, we restrict ourselves to the analysis of the case of a strongly magnetized plasma:

$$H = \frac{\omega_{ce}}{\nu_{ei}} \geq 1. \tag{9}$$

This mode is typical of experimental conditions [11], where the Hall parameter varies in the range of $10^3$–$10^4$. Then, when condition (9) is satisfied, the substitution of (8) into (7) gives the following:

$$m_i n \frac{dv_i}{dt} = \frac{1}{c}j \times B_0. \tag{10}$$

Thus, Equations (6), (8) and (10) completely describe the acceleration of a plasma bunch under the assumption that the plasma inside the bunch is homogeneous and quasi-neutral. This approximation is valid for the case when the Debye radius $\lambda_D$ is much smaller than the characteristic size between the particles in the plasma bunch and the absence of charge polarization, which occurs due to the drift of charged particles in crossed magnetic and electric fields. As a characteristic spatial scale of the plasma ring, one can take the ring length (see Figure 2), and then the condition of spatial homogeneity of the plasma ring can be written as $\lambda_D \ll \delta$. Herewith, according to [12], in the axial magnetic field $B_{z0}$ and the azimuthal electric field $E_\theta$ in the radial direction, there will be a separation of charges by the amount

$$\Delta R = \frac{(m_e + m_i)c^2 |E_\theta|}{eB_{z0}^2} \approx \frac{m_i c\, R_2}{2eB_{z0}^2}\left|\frac{\partial B_{z0}}{\partial t}\right|.$$

Since in the case under consideration $\Delta R \ll R_2$ must be satisfied, from this relation, we obtain the condition

$$\frac{m_i c}{2eB_{z0}^2}\left|\frac{\partial B_{z0}}{\partial t}\right| \ll 1, \tag{11}$$

making it possible to ignore the polarization of charges in crossed fields; i.e., we can neglect the internal electric and magnetic fields of the plasma ring. As a result, the bunch dynamics may be considered by taking into account only external fields $B_0$ and $E_0$ given by the relations (1) and (4).

In this case, for the magnetic field (1), we have

$$\nabla \times B_0 \equiv \left(\frac{\partial B_{r0}}{\partial z} - \frac{\partial B_{z0}}{\partial r}\right)e_\theta. \tag{12}$$

Bearing in mind relations (12) and (4) from the equation

$$j = \frac{c}{4\pi}\nabla \times B - \frac{1}{4\pi}\frac{\partial E}{\partial t} \tag{13}$$

we come to the conclusion that in the scheme under consideration,

$$j_z = J_r \equiv 0, \tag{14}$$

but $j_\theta \neq 0$. Since acceleration occurs under conditions of quasi-neutrality, it follows that for the velocity components of electrons and ions, we have $v_{ez} = v_{iz} = v_z$ and $v_{er} = v_{ir} = v_r$.

Now we can find $v_r$ and $v_z$: proceeding from Equation (10) and using condition (14) in this equation we obtain the following:

$$m_i \frac{\partial v_r}{\partial t} = \frac{e^2 B_{z0}}{cm_e \nu_{ei}} E_\theta + \frac{e^2 B_{z0}}{c^2 m_e \nu_{ei}} (B_{r0} v_z - B_{z0} v_r), \tag{15}$$

$$m_i \frac{\partial v_z}{\partial t} = -\frac{e^2 B_{r0}}{cm_e \nu_{ei}} E_\theta + \frac{e^2 B_{r0}}{c^2 m_e \nu_{ei}} (B_{z0} v_r - B_{r0} v_z). \tag{16}$$

At this stage, it should be noted that owing to the form of expression (8), the density $n$ drops out from Equations (15) and (16). In addition, from these equations, it follows that

$$B_{r0} \frac{dv_r}{dt} + B_{z0} \frac{dv_z}{dt} = 0. \tag{17}$$

By integrating (17) with respect to $t$ under the condition $v_r(t = 0) = 0$, we obtain

$$v_r = -\int_0^t \frac{B_{z0}}{B_{r0}} \frac{dv_z}{dt'} dt'. \tag{18}$$

Substituting (4) and (18) into Equation (16), we obtain

$$\frac{dv_z}{dt} = -\frac{e^2 B_{r0}}{cm_e m_i \nu_{ei}} E_\theta - \frac{e^2 B_{z0}}{c^2 m_e m_i \nu_{ei}} \int_0^t B_{z0} \frac{dv_z}{dt'} dt' - \frac{e^2 B_{r0}^2}{c^2 m_e m_i \nu_{ei}} v_z. \tag{19}$$

For simplicity, we shall neglect the radial dependence for $B_{z0}$, setting $B_{z0} = B_{z0}(t, z)$ in the gap $R_1 \leq r \leq R_c$. Then from (4) we obtain

$$E_\theta = -\frac{2r}{c} \frac{dB_{z0}}{dt}. \tag{20}$$

In addition, we restrict our consideration by the case

$$\frac{\partial^2 B_{z0}}{\partial t^2} > 0,$$

and then we have the following estimation:

$$\int_0^t B_{z0} \frac{dv_z}{dt'} dt' = \int B_{z0} dv_z \leq v_z(t) B_{z0}(t). \tag{21}$$

Taking into account relations (20) and (21) in (19), we arrive at

$$\frac{dv_z}{dt} \geq \frac{e^2 B_{r0} r}{2c^2 m_e m_i \nu_{ei}} \frac{dB_{z0}}{dt} - \frac{e^2 (B_{r0}^2 + B_{z0}^2)}{c^2 m_e m_i \nu_{ei}} v_z. \tag{22}$$

Now we shall go over dimensionless variables, namely

$$v = \frac{2v_z}{\omega_{ci} L}, \quad x = \frac{r}{L}, \quad z = \frac{z}{L}, \quad \tau = \omega_{ci} t, \quad \mathcal{B}_{r0} = \frac{B_{r0}}{B_0}, \quad \mathcal{B}_{z0} = \frac{B_{z0}}{B_0},$$

here $\omega_{ci} = eB_0/cm_i$ is the ion cyclotron frequency. In this case, bearing in mind the dependences (2) and (3) with condition (5), relation (22) can be rewritten as

$$\frac{dv}{d\tau} \geq H \frac{d\mathcal{B}_{z0}}{d\tau} - \frac{H}{x^2} v. \tag{23}$$

Now, instead of the inequality (22), we shall consider the corresponding equation:

$$\frac{dv}{d\tau} + \frac{H}{x^2} v = H \frac{d\mathcal{B}_{z0}}{d\tau}. \tag{24}$$

The general solution of this equation is

$$v(\tau) = \exp\left(-\frac{H}{x^2}\tau\right)\left[v_0 + H \int_0^\tau \frac{d\mathcal{B}_{z0}}{d\tau'} \exp\left(\frac{H}{x^2}\tau'\right) d\tau'\right], \tag{25}$$

where $v_0 = v(\tau = 0)$. According to the Chaplygin comparison theorem [13], the solution of this linear differential equation can be treated as an estimate majorizing Equation (23) from below. In other words, this characteristic lower estimation determines the characteristic behavior of the axial velocity component for the worked-out scheme.

## 4. A Practical Example

As is seen from (25), the acceleration dynamics is mainly determined by the Hall parameter $H$ and a time-varying axial magnetic field $\mathcal{B}_{z0}$ in the solenoid. As a typical experimental dependence of an axial magnetic field, one can take

$$\mathcal{B}_{z0}(\tau) = \beta \frac{\tau}{\tau_i} \exp\left(1 - \frac{\tau}{\tau_i}\right), \tag{26}$$

where $\tau_i$ is the dimensionless duration of pulse with the specified magnitude (see Equation (3)) relating to the actual pulse duration $t_i$, and the relative amplitude $\beta = B_s/B_m$ displays the ratio between the axial and radial components of the magnetic field. As previously mentioned, according to relation (2), such a magnetic field can be generated by an electric pulse of the same shape and duration $t_i$ having some amplitude $I_0$. Substituting (26) into (25), we obtain

$$v(\tau) = v_0 \exp\left(-\rho \frac{\tau}{\tau_i}\right) + \frac{\beta H}{(1-\rho)^2}[\rho + \Omega\tau - \rho \exp(\Omega\tau)] \exp\left(1 - \frac{\tau}{\tau_i}\right), \tag{27}$$

where the following useful parameters are used:

$$\rho = \frac{H\tau_i}{x^2}, \quad \Omega = \frac{1}{\tau_i} - \frac{H}{x^2} = \frac{1}{\tau_i}(1-\rho). \tag{28}$$

It is noteworthy that the physical conditions required for the implementation of the discussed acceleration model are routine for the experiments [11,14] and close to the schematic presentation shown in Figure 1. That is, the present model may be useful for a preliminary analysis of the performance of various experimental devices of the type under consideration. So, to fully display this point, we perform some estimations for typical experimental conditions [11,14]. Since these experiments were carried out with the magnetized plasma of density $n_e \approx 9 \times 10^{12}$ cm$^{-3}$ and average electron temperature $T_e \approx 1$ eV, one can set $\ln(\Lambda) = 5$. The characteristic value of the radial magnetic field was $B_m = 50$ G. For these parameters, we have $H = 10$. As typical parameters of the experimental system, we take the current pulse with amplitude $I_0$ for the interval $t_i = 5 \times 10^{-7}$ s passing in the solenoid with $k = N/h_c = 10$ coils per centimeter, which gives $t_i > \omega_{ci}^{-1}$. Then, using these values in the definition $B_s$ we obtain $B_s = 12 I_0$ where, for clarity, the amplitude $B_s$ is expressed in gauss, and the current $I_0$ is recorded in amperes. Inserting this relation into (5), we obtain the inequality $I_0 < B_m/12$ which defines the

admissible $I_0$ for the current value $B_m$. For example, for the present $B_m$, one can take $I_0 = 2$ A, which yields $B_s = 24$ G. As one can see, these values are available experimental parameters, and the relations (9), (11) and (26) are fulfilled.

Therefore, using relation (27), we pass over to the consideration of the dynamics of the plasma bunch in typical experimental conditions. For an initial velocity $v_0$, we take the ion sound speed $c_S = \sqrt{T_e/m_i}$, which is the characteristic velocity at the boundary of the plasma layer. For the convenience of presentation, Figure 3 presents the dependence for axial velocity $v$ normalized to the initial plasma velocity $v_0$ as a function of $\tau/\tau_i$ for different radius $x$ values. We see that the graphs for different $x$ values are alike in general. In all these cases, we observe a significant increase in the axial velocity (approximately by a factor of 60–80) in the interval $0 \leq \tau \leq 0.5\,\tau_i$. This means that the discussed schematic has significant technical potential.

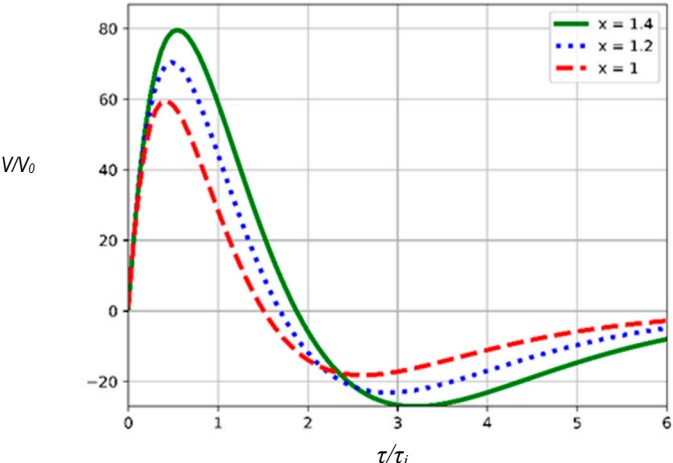

**Figure 3.** The dependence of $v/v_0$ as a function of $\tau/\tau_i$ for the different values of dimensionless radius $x$.

As is seen from these graphs, increasing radius $x$ via parameter $\rho$ (see Equation (28)) results in an even greater divergence of the curves for times $\tau \geq 0.5\,\tau_i$. In fact, such a process can lead to the violation of the integrity of the plasma ring. These results show that there exists a small interval in the system parameters in which it is possible to see the effects occurring in reality. So, the model worked out may be useful for the advanced analysis of plasma acceleration in some limited regions of system parameters.

## 5. Concluding Remarks

In this paper, we have considered the design of a plasma thruster that accelerates a plasma flow in the crossed magnetic fields. The key point of the study is to show the acceleration effect in the proposed design and its technical implementation in the simplest way. The presented relations (2), (9) and (11) define the range of allowable parameters of the device where one can obtain an acceleration effect. Proceeding from all these points, we can outline the directions for further development of the proposed acceleration scheme.

In the case under consideration, the acceleration effect is caused by the azimuthal electric field which comes from the temporal variation of axial magnetic flux (3) in the cylindrical solenoid volume where $\mu = 1$ (see Figure 1). The interaction of azimuthal electron and ion flows with the radial component $B_{r0}$ (see Equation (2)) via the Lorentz force brings about the axial acceleration of the plasma flow as a whole. Thus, the acceleration effect is proportional to the product $B_{r0}\partial_t B_{z0}$ but does not depend on the magnitude of $B_{z0}$, and this opens up new opportunities. It is clear that the change in the cylindrical form of the coil and the use of special ferromagnetic inserts in the design can significantly change both the magnetic flux in the acceleration region and the radial component $B_{r0}$. Another technical opportunity is related to the current discharge organization in the solenoid that

directly determines the change in $B_{z0}$ with respect to time (see Equation (4)). That is, if the distribution $B_{z0}$ as a function of time changes faster than it would with the choice of Equation (26), we will obtain a larger azimuthal field. Herewith, in order to avoid breaking the integrity of the plasma ring, condition (11) must be satisfied, which restricts the choice of admissible dependence for $B_{z0}$. Thus, all these points must be taken into account to optimize the proposed accelerator scheme.

In conclusion, let us especially stress that our consideration was limited to the plasma flow consisting of only electrons and ions. However, proceeding from the results of [15,16], one can expect that the proposed acceleration method may be used for the acceleration of multicomponent plasma flows. This point should be useful in the design of plasma thrusters [17] and the studies of geophysical processes in near-Earth space [18,19].

We dedicate our research and this paper to the memory of Professor V.A. Kurnaev and Associate Fellow of the AIAA P.A. Murad. The authors would like to express their profound gratitude to the referees for the valuable remarks and suggestions on the lines of further research.

**Author Contributions:** Conceptualization, A.K. and V.Y.; methodology, A.K. and V.Y.; validation, A.K., S.T. and V.Y.; investigation, A.K., V.Y. and S.T.; writing review and editing, A.K. and V.Y.; supervision, A.K.; funding acquisition, A.K. All authors have read and agreed to the published version of the manuscript.

**Funding:** The work was financially supported by the Ministry of Science and Higher Education of the Russian Federation (agreement Nos. 075-15-2021-1361 and 075-01129-23-00).

**Institutional Review Board Statement:** Not applicable.

**Informed Consent Statement:** Not applicable.

**Data Availability Statement:** Not applicable.

**Conflicts of Interest:** The authors declare no conflict of interest.

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
