# Peer review of "Pulsed Plasma Accelerator"

_plasma, doi:10.3390/plasma6010004_

Round 1
Reviewer 1 Report
The paper reports the description of physical acceleration of plasma charged particles due to a particular electromagnetic configuration, which allows to transfer longitudinal force of not stationary magnetic fields to electrons and ions.
The study started from the original assumption well defined and developed by the authors in jobs previously published and in this context the proof of principle of his application was presented.
Despite the rigorous analytical approach, however the beam dynamics of plasma interaction with EM fields and the related particles acceleration must be faced in an advance way, i.e. by means of particle-in-cell based codes.
My strong suggestion is to pursuit this kind of method to finalize the beam dynamics and present the results in next published job.
Finally, the present work can be published after Author will define the Debye relation reported at pag 3.
Reviewer 2 Report
The manuscript is rather close to a textbook chapter than to a research article. It contains a lot of mathematical formula which are difficult for the readers to follow. The abstract is not self-sufficient since it refers to tow publications, this is unusual and not conventional. The authors should explain more clearly the problem they try to solve and why it is intresting to solve, this will allow to reach a more general audience. A conclusion is also missing. There also some mistakes in the rest of the text. For instance in caption of Fig. 1, ref 8 should be replaced by ref 9 like in page 2 (where ref 9 is cited). Even if it can be guessed, the coordinate system is not explicitely indicated (I guess cylindrical system because of cylindrical symmetry).
After eq (4), what is meant by "performing two functions" ?
There is an error in the formula of the cycltron frequency where the speed of light c is incorrectly introduced. Many symbols are not explaind like delta and lambda_D.
In page 4, just after eq (13) the authors may indicate that the different v represent velocities.
In eq(17) what does the star mean in dt*? same in eq (21).
In page 6, eq (25) what does y stand for? what is its dimension?
Just after eq (26) the authors say that tau_n is dimensionless but in the second but last line a value of tau_n is expressed in seconds.
In page 7, x-label of fig 2 is tau/tau_H while in the caption a ratio tau/tau_n is used. What is correct tau_H or tau_n?
Replace 1R_1 by R_1.
The manuscript should be revised
Round 2
Reviewer 2 Report
The authors have improved their manuscript and clarified many points.I think that the manuscript can be published in the present version even tough some improvements are still possible. I noticed some minor errors that can be corrected before the proof edition. These are detailed below:
- Page 4: "on the itself magnitude"--> remove "itself".
- Page 6: "describing by the relations (1) and (4)" --> given by relations (1) and (4).
- No need to use "the" before relations (12)...
- Page 8:
In page (7) the parameter tau=omega_ci times t is a dimensionless time. So is tau_i after relation (26). In my understanding duration t_i is such that tau_i = omega_ci times t_i. You should indicate this since t_i is used and a value is given later.
- according (2) --> according to (2) or according to relation (2).
- t_i times omega_ci >1 means tau_i>1. Is it right?
- Page 9.
Ok even if you are using cgs units and x to designate radial coordinate, the x values unit is missing, I guess cm (x=1.4 cm, x=1 cm, x=1.2 cm) in the figure.
- Page 10: "but the one does not depend ...opens up new opportunities." --> "but does not depend on the magnitude of B_z0 and this opens up new opportunities.".
- the condition (11) --> condition (11).
No need for me to check another version.
Author Response
We again thank the referee for careful reading of our manuscript and useful comments. New changes are marked in green color.
We hope that our manuscript is now suitable for publication.
